# Fair Image Generation from Pre-trained Models by Probabilistic Modeling

**Mahdi Ahmadi, John Leland, Agneet Chatterjee & YooJung Choi**
School of Computing and Augmented Intelligence
Arizona State University
Tempe, AZ 85281, USA
{ahmadi,jslelan1,agneet,yj.choi}@asu.edu

## Abstract

The production of high-fidelity images by generative models has been transformative to the space of artificial intelligence. Yet, while the generated images are of high quality, the images tend to mirror biases present in the dataset they are trained on. While there has been an influx of work to tackle fair ML broadly, existing works on fair image generation typically rely on modifying the model architecture or fine-tuning an existing generative model which requires costly retraining time. In this paper, we use a family of tractable probabilistic models called probabilistic circuits (PCs), which can be equipped to a pre-trained generative model to produce fair images without retraining. We show that for a given pre-trained generative model, our method only requires a small fair reference dataset to train the PC, removing the need to collect a large (fair) dataset to retrain the generative model. Our experimental results show that our proposed method can achieve a balance between training resources and ensuring fairness and quality of generated images.

## 1 Introduction

In recent years, generative models have seen an explosion of interest and advancements and are being applied in a wide range of real-world domains across different modalities including image, text, etc. Some popular examples of image-based generative models are variational autoencoders (VAEs) (Kingma, 2013), generative adversarial networks (GANs) (Goodfellow et al., 2020), flow-based generative models (Dinh et al., 2014), and diffusion models (Sohl-Dickstein et al., 2015). Similarly, text-based generative models such as GPT-4 (Achiam et al., 2023) , Bloom (Le Scao et al., 2023), Gemini (Team et al., 2023), and LLAMA (Touvron et al., 2023) have advanced to more reliably produce high quality text.

However, while generative models have demonstrated success across many domains in producing realistic samples, fair generation has been relatively underexplored. Fair generation is a difficult challenge as it is a direct consequence of bias in the large amount of data needed to train generative models. Yet, as generative models become integrated into everyday life, it is paramount to establish techniques to ensure fair generation.

One of the first challenges to address in fairness-aware machine learning is determining the appropriate notion of fairness for the domain. For example, for fair classification, there exists a number of technical formulations of fairness, often concerning the output (prediction) of the classifier and sensitive/protected attributes such as race, gender, or other demographic features. On the other hand, for image generation, there is no particular output/target variable with respect to which to measure fairness. Rather, the users and ML practitioners for downstream tasks may be interested in the distribution of the generated samples and ensuring that it is not biased with respect to some sensitive attributes (Sattigeri et al., 2019). For instance, this notion of fairness could simply enforce that the sensitive attributes of the generated images follow a uniform distribution (e.g., equal probability of generating a female or male image); however, it may be more appropriate in certain domains to rather enforce that the generated distribution matches the population distribution. As such, this

work focuses on ensuring that the distribution of generated images follows a given reference fair distribution.

This paper makes the following key contributions:

- We propose to use probabilistic circuits (PCs) to learn the distribution of a reference fair dataset. This can be done for both cases of reference dataset with or without sensitive attribute information. In the case where sensitive attributes are available, we also show that we can leverage this to make the sampling distribution fair post-hoc.

- Our method can be integrated with a variety of pre-trained generative models: we only assume that there exists an encoder to map images to some lower-dimensional latent representation.

- We empirically show that both proposed methods have much shorter training times while generating fairer images than the base generative model.

## 2 RELATED WORK

Discussions regarding fair image generation have grown in the machine-learning community due to the increased utilization of generative models. Previous approaches to fair image generation involved transfer learning (Teo et al., 2023), generating fair synthetic data (Van Breugel et al., 2021), and learning without sensitive labels (Um & Suh, 2023; Jalal et al., 2021). Pioneering work in this area includes (Choi et al., 2020a), which uses a weakly-supervised approach to learn a generative model based on an importance reweighing scheme. Most related to our work is Tan et al. (2020) which also tackles the problem of fair image generation, specifically focusing on generative adversarial networks (GANs) (Goodfellow et al., 2020) based on an empirical finding that bias within training data is magnified by the GAN model. They propose the use of "latent distribution shifting" by learning a Gaussian mixture model (GMM) over a set of fair latent codes conditioned on a specific attribute value.

In this work, we utilize a more general and expressive model (probabilistic circuits) to closely capture the distribution of a reference fair dataset. This allows us to integrate directly into a variety of pre-trained generative models similar to (Tan et al., 2020). However, the increase of expressivity allows us to easily represent more complex latent distributions (both discrete and continuous) over the fair subset of attributes.

## 3 BACKGROUND

**Notation** Random variables are denoted by uppercase letters $(X)$ and their assignments by lowercase letters $(x)$. We use bold uppercase $(\boldsymbol{X})$ and lowercase letters $(\boldsymbol{x})$ for sets of variables and their assignments, respectively. $Pr$ represents probability, $p$ and $q$ represent probability mass functions, and $\mathbb{E}[.]$ represents expected value. We use $\boldsymbol{S}$ to denote a set of sensitive attributes (e.g., gender, race, etc.) that we wish to be fair with respect to. For a given sample $\boldsymbol{x}$, we denote its associated sensitive attribute information by $\boldsymbol{s}(\boldsymbol{x})$.

### 3.1 FAIR GENERATION

Suppose $\boldsymbol{S}_{\text{fair}}$ is the distribution of the sensitive attributes that we wish our generated samples to match. That is, the goal is to have a model capable of generating samples with distribution $p_{\text{fair}}(\boldsymbol{x})$ where $\mathbb{E}_{\boldsymbol{x} \sim p_{\text{fair}}}[\boldsymbol{s}(\boldsymbol{x})] = \boldsymbol{S}_{\text{fair}}$. The reference distribution may, but is not required to, follow a uniform distribution with respect to the sensitive attributes. For example, in terms of gender equality, usually $s \sim \text{Bernoulli}(0.5)$, meaning that the gender of a generated sample has equal probability for female and male. Generative models need a large dataset (denoted $\mathcal{D}_{bias}$) to produce high-quality images, and may mirror the biases of the dataset. However, the biases in the learned model are usually to the detriment of the its performance across minority groups. Suppose that we have access to a (usually small) reference dataset, called $\mathcal{D}_{fair}$, whose sensitive attributes follow the distribution $\boldsymbol{S}_{\text{fair}}$. The goal in fair generation is to use both $\mathcal{D}_{fair}$ and $\mathcal{D}_{bias}$ to train a generative model that is both expressive and fair.

### 3.2 PROBABILISTIC CIRCUITS

Probabilistic Circuits (PCs) (Choi et al., 2020b) are a family of tractable probabilistic models (TPMs) such as sum-product networks (Poon & Domingos, 2011), Einsum networks (Einets) (Peharz et al., 2020), Cutset networks (Rahman et al., 2014), and arithmetic circuits (Darwiche, 2002; 2003). Here, tractability refers to the ability to compute probabilistic queries in polynomial time, such as computing marginal or conditional probabilities. Probabilistic circuits can also be seen as deep mixtures of simple probability distributions.

**Definition 1** (Probabilistic Circuits) (Dang et al., 2022) A PC, $\mathcal{C} := (\mathcal{G}, \theta)$, represents a joint probability distribution $p(\boldsymbol{X})$ over random variables $\boldsymbol{X}$ through a directed acyclic graph (DAG), $\mathcal{G}$, which is parameterized by $\theta$. The DAG is composed of 3 types of nodes: leaf, product $\otimes$, and sum $\oplus$. Every leaf node in $\mathcal{G}$ is an input, and every inner node receives inputs from its children $in(n)$. Every node computes a probability density function $p(\boldsymbol{X})$, defined recursively as follows:

$$
p_n(\boldsymbol{x}) := \begin{cases} f_n(\boldsymbol{x}) & \text{if } n \text{ is a leaf,} \\ \prod_{c \in in(n)} p_c(\boldsymbol{x}) & \text{if } n = \otimes, \\ \sum_{c \in in(n)} \theta_{c|n} p_c(\boldsymbol{x}) & \text{if } n = \oplus, \end{cases} \tag{1}
$$

where $f_n(x)$ is some univariate input distribution, $\theta_{c|n}$ is the parameter associated with the edge connecting nodes $(n, c)$ in the graph $\mathcal{G}$ and $\sum_{c \in in(n)} \theta_{c|n} = 1$. The probability distribution represented by a circuit is that of its root: $p_{\text{root}}(\boldsymbol{X})$. Throughout this paper, we will focus on categorical random variables at the leaves with inputs $x \in \{0, \dots, K-1\}$ where $K$ is the number of categories.

Based on this recursive definition, joint likelihoods can easily be evaluated from the circuit using a single "feed-forward" pass through the circuit from leaf to root. However, to ensure tractable inference, we must ensure additional circuit structural properties; most notably, for tractable marginal and conditional inference, we need the circuit to be *smooth* and *decomposable* (Choi et al., 2020b).

**Definition 2** (Smoothness and Decomposability) The scope of a node in a PC, $n$, is the set of input variables to $n$. We refer to a product node as *decomposable* if its children have disjoint scopes, and refer to a sum node as *smooth* if its children have identical scopes. A PC is called smooth and decomposable iff all of its sum nodes are smooth and all of its product nodes are decomposable.

Throughout this paper, we will assume that all PCs used are smooth and decomposable, giving us the ability to compute not just marginals in linear time, but also to conditionally sample from a PC in linear time. Furthermore, for notational simplicity, we assume without loss of generality that the circuit has alternating sum and product nodes. An example of a simplified PC structure which we will use throughout this work can be seen in Figure 2.

## 4 PROPOSED METHOD

Assuming that there exists a pre-trained generative model having some kind of encoder-decoder pair—e.g., an autoencoder or a flow-based model, etc.—we use a probabilistic model to intervene on its latent distribution. That is, we will guide the latent variables such that the images generated by decoding those latent variables will follow a fairer distribution with respect to a reference dataset. In different image generation paradigms, one may also work with a generator such as a GAN, as seen in (Tan et al., 2020). The details regarding the specific choice of model for our implementation are provided in Section 5. According to Figure 1, the encoder can be modeled by $q_\phi(\boldsymbol{z}|\boldsymbol{x})$ where $\boldsymbol{x}$ is the input image, $\boldsymbol{z}$ is the latent vector, aka embedding, and $\phi$ represents all the parameters in the encoder. Similarly, the decoder is modeled by $p_\theta(\tilde{\boldsymbol{x}}|\boldsymbol{z})$ where $\tilde{\boldsymbol{x}}$ denotes the reconstructed image, and $\theta$ the decoder parameters.

Probabilistic circuits have demonstrated strong capabilities in expressively modeling distributions while preserving tractability. It gives them the ability to perform marginal and conditional queries in polynomial time with respect to the circuit size. Probabilistic circuits are also model-efficient; they have a relatively smaller number of parameters than NN-based generative models. In this work, a PC learns the distribution of latent variables for $\mathcal{D}_{fair}$. In other words, a PC parameterized by $\psi$ is learned, i.e. $p_\psi(\boldsymbol{z}) = Pr(\boldsymbol{Z} = \boldsymbol{z}; \psi)$. Then with the help of the decoder, the distribution of the

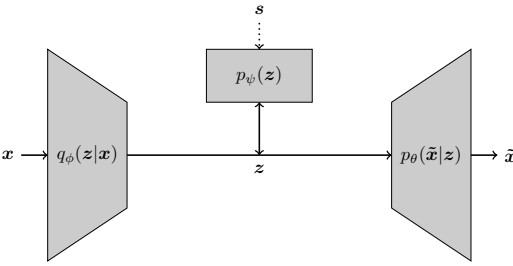

Figure 1: Given an encoder-decoder pair, a probabilistic circuit parameterized by $\psi$ learns the distribution of latent variable $\boldsymbol{Z}$. The sensitive attributes $\boldsymbol{S}$ can be used jointly with $\boldsymbol{Z}$ to improve the training.

---

**Algorithm 1** Learning a PC and sampling images according to it

1: **Input:** Trained encoder $\mathcal{E}n$ and decoder $\mathcal{D}e$ pair, PC $\mathcal{C}$, Large biased dataset $\mathcal{D}_{bias}$, Reference fair dataset $\mathcal{D}_{fair}$.
2: **Output:** Fair image samples $\tilde{\boldsymbol{X}}_{gen}$
3: Encode $\mathcal{D}_{fair}$ training split to $\tilde{\boldsymbol{Z}}_{fair}$ with $\mathcal{E}n$;
4: Train $\mathcal{C}$ with input=concat($\boldsymbol{Z}_{fair}, \boldsymbol{S}$) according to Algorithm 2
5: Sample new latent variables $\boldsymbol{Z}_{gen}$ from $\mathcal{C}$ according to Algorithm 3
6: Decode $\boldsymbol{Z}_{gen}$ to $\tilde{\boldsymbol{X}}_{gen}$ by $\mathcal{D}e$
7: **return** $\tilde{\boldsymbol{X}}_{gen}$

---

output, $\tilde{\boldsymbol{x}}$, should be fair. Given this formulation requires us to only adjust the latent variables and does not require us to fine-tune the generative model, the proposed method is both theoretically and experimentally seen to be faster than competing methods.

Given an ideal encoder and decoder pair, meaning that they are not affected by the biases in $\mathcal{D}_{bias}$, one naive approach is to learn the distributions of the latent variables when the fair dataset ($\mathcal{D}_{fair}$) is fed to the encoder (see Figure 1). Considering this scenario, a PC learns the latent space distribution i.e. $p_{fair}(\boldsymbol{z})$. In our learning procedure, we use SGD-based negative likelihood loss defined as $\mathcal{NLL} = -\sum_{\boldsymbol{z} \in Batch} \log(p(\boldsymbol{z}))$. The algorithm for learning with negative likelihood is left in Appendix 2.

While this approach seems to be promising, we will show in the experiments (Section 5) that this method does not generate samples with a satisfactory level of fairness. Note that in this case, the PC does not have access to the sensitive attributes $S$. The problem with this approach is that the latent variables ($\boldsymbol{z}$) do not follow the fair distribution even when $\boldsymbol{x} \sim p_{fair}(\boldsymbol{x})$ because the encoder tends to skew their output distributions toward the majority group, i.e., $p_{encoder}(\boldsymbol{z}) = \mathbb{E}_{\boldsymbol{x} \sim p_{fair}(\boldsymbol{x})} Pr(\boldsymbol{Z} = \boldsymbol{z}|\boldsymbol{x}; \phi) \neq p_{fair}(\boldsymbol{z})$. This issue is magnified when the decoder similarly skews towards the majority group, even if it is over a set of fair latent variables, that is, $p_{decoder}(\tilde{\boldsymbol{x}}) = \mathbb{E}_{\boldsymbol{z} \sim p_{fair}(\boldsymbol{z})} Pr(\tilde{\boldsymbol{X}} = \tilde{\boldsymbol{x}}|\boldsymbol{z}; \theta) \neq p_{fair}(\tilde{\boldsymbol{x}})$.

So, we propose an alternative to resolve the aforementioned issue. You can see an overview of the proposed method in Algorithm 1. In case we also have access to the sensitive attributes of the fair dataset, we can train a PC on the joint distribution of $\boldsymbol{S}$ and $\boldsymbol{Z}$ such that we learn $p_{fair}(\boldsymbol{z}, \boldsymbol{s})$. We can call the new approach guided learning as opposed to the previous unguided method. To do this, we construct a PC with two identical sub-circuits. They are connected to a sum node by conditioning on $\boldsymbol{S}$. A simplified version of the proposed structure with only two latent variables is shown in Fig. 2. The training algorithm is presented in Alg. 2. It is the same as the case not having access to $\boldsymbol{S}$. The only difference is using concat($\boldsymbol{Z}, \boldsymbol{S}$) instead of $\boldsymbol{Z}$.

This can be viewed from another perspective. With a noisy latent variable for the minority group, say $\boldsymbol{Z} + \boldsymbol{n}$, the distribution has more variance and, therefore, less density. So, the overall circuit assigns a higher relative weight ($w_S$ and $w_{\neg S}$) to compensate for its lack of probability density. In sampling time, each subcircuit can contribute to the sampling process based on its relative weight (the sampling algorithm is provided in the Appendix). All in all, the subcircuit for the under-represented

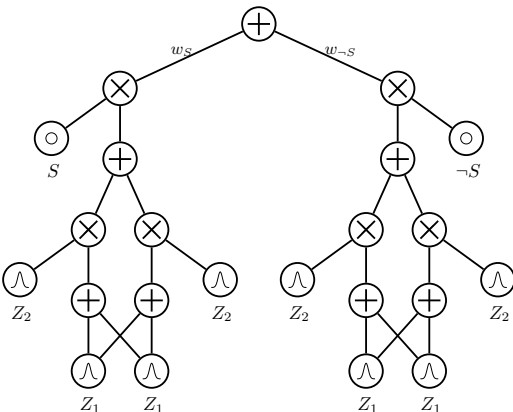

Figure 2: A simplified sketch of the proposed PC structure for distribution with one binary sensitive attribute. Two sub-circuits with identical structures are connected to product nodes controlled by $\boldsymbol{S}$, then a sum node connects the resulting circuits.

group will contribute more to the sampling process. Note that we do not specify the sensitive attribute in sampling time, and it is determined by the sampling algorithm itself.

Furthermore, with a PC representing such joint distribution, our method can also support sampling according to an arbitrarily specified sensitive attribute distribution. In particular, this is thanks to the tractable conditioning and sampling capabilities of PCs, as well as their parameters having interpretable semantics (e.g., $w_s$ and $w_{\neg s}$ in Figure 2 corresponding to the probability of $s$ and $\neg s$, respectively). For example, we can conditionally sample according to arbitrary sensitive attribute distribution $\boldsymbol{S}$ shown by $p(\boldsymbol{z}|\boldsymbol{s} = \boldsymbol{s_0})$ in which $\boldsymbol{s_0}$ is an assignment to $\boldsymbol{s}$; alternatively, we can entirely fix the distribution by modifying the corresponding circuit weights accordingly. However, in our experiments, we observed that this approach depends heavily on the reliability of the pre-trained encoder-decoder pair to accurately reproduce images of particular sensitive attributes.

## 5 EXPERIMENTS

### 5.1 EXPERIMENTAL SETUP

Each experiment setting was repeated 10 times, and the average result is reported. For each run, 10,000 samples are generated. Tolerance $\epsilon$ (see Algorithm 2) is set to 1, and the maximum number of epochs and batch size were set to 2000 and 2048, respectively. We used an NVIDIA L40S GPU with 48GB of memory for the experiments.

In this paper, we use a variation of GANs called vector-quantized GAN, or simply VQ-GAN (Esser et al., 2020), as it is shown to be able to generate high-quality samples. We work only with its encoder and decoder and not its transformer on the discriminator. The idea underlying these models is to quantize the latent vectors to their nearest code-book vector, resulting in an integer number (code-book index). More accurately, an $M \times N \times R$ float embedding tensor will be converted to an $M \times N \times 1$ integer tensor, where $M$ and $N$ are the spatial and $R$ is the channel dimensions.

We use Einsum networks (Peharz et al., 2020) as the building block of our proposed PC structure. The network has a depth of 3, with the number of sums, leaves, and repetitions all set to 10. The number of categories in the VQ-GAN code book is set to 1024, so we use the Einsum network with categorical leaf nodes of the same size.

All experiments were performed on the CelebA dataset (Liu et al., 2015). The images have dimension $64 \times 64$, resulting in latent variable dimensions of $|\boldsymbol{Z}| = 8 \times 8 = 64$ after encoding. The selected sensitive attribute for all the experiments is the gender of the face image, and the fair female-male ratio is selected to be 50-50. Note that the gender of the images is only used in the experiments with guided learning (shown in figures and tables by "Ours + $\boldsymbol{S}$".)

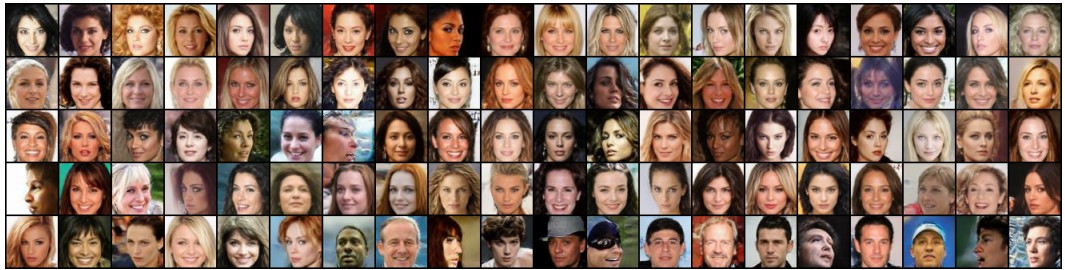

Figure 3: Generated samples using VQ-GAN (Esser et al., 2020) transformer when the female-male ratio in $\mathcal{D}_{bias}$ is 90-10, and $\gamma = 0.25$. For this set of generated samples, the proportion of females is 0.86 and males is 0.14

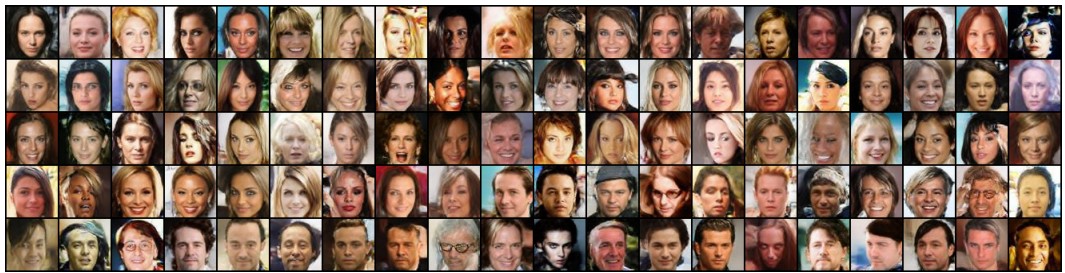

Figure 4: Generated samples by the first proposed method, i.e., unguided learning when the female-male ratio in $\mathcal{D}_{bias}$ is 90-10, and $\gamma = 0.25$. For this set of generated samples, the proportion of females is 0.64 and males is 0.36

Instead of using two different datasets, CelebA is divided into $\mathcal{D}_{bias}$ and $\mathcal{D}_{fair}$. The unfair dataset ($\mathcal{D}_{bias}$) can have different degrees of bias/unfairness (we refer to it as F-M Ratio.) In addition, the relative size of these two datasets is important and is referred to as $\gamma$. The generative model is trained on the unfair dataset. However, the PC is trained on the fair subset's latent variables passed to that trained generative model.

## 5.2 RESULTS

The proposed method is evaluated by different metrics measuring the quality and fairness of the samples. The first metric is the total variation distance (TVD) between the sensitive attribute distributions of the generated images and the reference dataset. We refer to it as fairness discrepancy (FD). It is computed by:

$$\text{FD} = \text{TVD}(p_{out}, p_{fair}) = \frac{1}{2} \sum_{\boldsymbol{s}} |p_{out}(\boldsymbol{s}) - p_{fair}(\boldsymbol{s})| \tag{2}$$

where $p_{out}$ is the distribution of sensitive attributes for the generated images and $p_{fair}$ is its distribution in the reference fair dataset ($\mathcal{D}_{fair}$). We use a classifier trained on the original CelebA images to predict the sensitive attributes of the generated images. We use the same ResNet-18 (He et al., 2016) classifier as used in (Choi et al., 2020a). Note that this classifier is only for metric purposes and is not used during training or sampling.

The samples generated from VQ-GAN (Esser et al., 2020) transformer for one set of dataset configurations, i.e., bias = 90-10, and the dataset ratio $\gamma = 0.25$. can be found in Fig. 3. In this case, we utilized the VQ-GAN transformer to produce the latent variables. We set the temperature to 1.0 and $k$ in top-k sampling to 600. The results for the same dataset configurations when sampling from our proposed structure first methodare presented in Fig. 4. As can be seen, there are more male samples in our method than using just VQ-GAN. Similarly, the results for second method ($\boldsymbol{Z} + \boldsymbol{S}$)are presented in Fig. 5. According to the results, the samples are fairer than the first method.

According to the experiments, when a dataset has greater bias, the effectiveness of the PC to represent the fair distribution is diminished. This is because the generative model is biased, so its latent

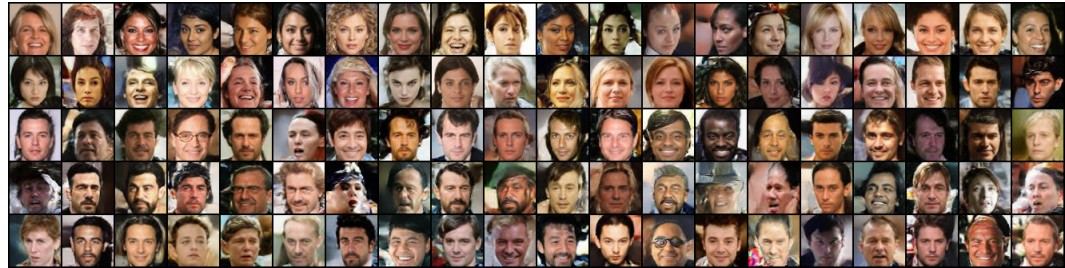

Figure 5: Generated samples by the second proposed method, i.e. guided learning, when female-male bias in $\mathcal{D}_{bias}$ is 90-10, and $\gamma = 0.25$. For this set of generated samples, the proportion of females is 0.42 and males is 0.58

Table 1: FD, FID, and inception scores (IS) for the proposed method and (Choi et al., 2020a) and (Esser et al., 2020). The results are presented for different configurations of $\mathcal{D}_{bias}$.

| F-M ratio | | 80-20 | | | 90-10 | | |
|---|---|---|---|---|---|---|---|
| | $\gamma$ | FD($\downarrow$) | FID($\downarrow$) | IS($\uparrow$) | FD($\downarrow$) | FID($\downarrow$) | IS($\uparrow$) |
| Esser et al. (2020) | - | 0.273 | 24.13 | 2.024 | 0.354 | 22.03 | 2.005 |
| Choi et al. (2020a) | 0.1 | 0.500 | 414.48 | 1.033 | 0.077 | 307.39 | 1.381 |
| | 0.25 | 0.385 | 25.68 | 1.825 | 0.298 | 27.05 | 1.923 |
| | 0.5 | 0.316 | 20.98 | 2.028 | 0.350 | 23.26 | 1.960 |
| | 1.0 | 0.270 | 17.54 | 2.123 | 0.321 | 17.45 | 2.019 |
| Ours | 0.1 | 0.151 | 26.91 | 1.953 | 0.223 | 28.57 | 1.893 |
| | 0.25 | 0.147 | 26.19 | 1.939 | 0.164 | 24.46 | 1.939 |
| | 0.5 | 0.146 | 26.05 | 1.938 | 0.217 | 27.83 | 1.881 |
| | 1.0 | 0.144 | 26.08 | 1.938 | 0.214 | 27.30 | 1.873 |
| Ours $+S$ | 0.1 | 0.082 | 33.25 | 2.020 | 0.020 | 32.40 | 1.968 |
| | 0.25 | 0.133 | 34.98 | 1.970 | 0.119 | 32.38 | 1.951 |
| | 0.5 | 0.143 | 35.23 | 1.948 | 0.070 | 33.33 | 1.902 |
| | 1.0 | 0.150 | 36.12 | 1.954 | 0.070 | 33.45 | 1.901 |

variables tend to be more skewed toward the majority group. One experiment we did was to encode and decode the original images in $\mathcal{D}_{bias}$ (having uniform distribution for females and males) and then classify them. The female-male ratio of the reconstructed images was 57.46-42.53 when the autoencoder was trained with a bias of 90-10, and it was 56.45-43.54 when trained with a bias of 80-20. It is obvious from the numbers that even encoding and decoding the original images with the learned generative model tends to be classified toward the majority group. One direct consequence of this issue is that when performing conditional sampling, e.g., sampling from $p(\boldsymbol{z}|\boldsymbol{s})$, we didn't get satisfactory results. So, we will leave conditional sampling for future work.

The fairness discrepancy (FD) scores for the proposed method versus the one for (Choi et al., 2020a) are presented in Table 1. As can be seen, the proposed method has a better FD score for both of our implementations.

The quality of the generated images is measured by Fréchet inception distance (FID) (Heusel et al., 2017) and inception score (Salimans et al., 2016). The FID and inception scores for the proposed method and (Choi et al., 2020a) are also presented in Table 1. The results show that both the proposed methods are robust to changes in $\mathcal{D}_{fair}$ to $\mathcal{D}_{bias}$ ratios ($\gamma$). It also means that the proposed method is very data-efficient and can work with smaller reference dataset sizes.

The average training times can be found in Table 2. According to this table, the proposed method is one order of magnitude faster than the baseline.

Table 2: Average training time (in minutes) for the proposed method and (Choi et al., 2020a)

| F-M Ratio | 80-20 | | | | 90-10 | | | |
|---|---|---|---|---|---|---|---|---|
| $\gamma$ | 0.1 | 0.25 | 0.5 | 1.0 | 0.1 | 0.25 | 0.5 | 1.0 |
| Choi et al. (2020a) | 109.13 | 242.48 | 285.97 | 371.65 | 216.00 | 239.25 | 285.62 | 372.35 |
| Ours | 3.35 | 6.03 | 8.36 | 11.27 | 3.50 | 6.75 | 8.67 | 12.63 |
| Ours $+S$ | 5.13 | 10.88 | 14.41 | 18.66 | 5.12 | 11.28 | 15.27 | 22.50 |

## 6 CONCLUSION

In this work, we utilize the capabilities of probabilistic models to ensure the generation of fair images. More exactly, by using probabilistic circuits, the latent distribution of a fair reference data was learned without any need to fine-tune the generative model. We showed that the generated images have sufficient fidelity while following a fair distribution. We find experimentally that the proposed method is much faster and more data-efficient than the existing methods, as this method only requires an encoder to map from fair images to their latent variables. While our current implementation worked with VQ-GAN, the proposed method can in theory be used with other generative models such as flow-based models. We found that while methodologically unguided distribution learning is possible, it can result in the encoder skewing the latent variables to the majority group. To correct this issue, we shifted towards guided distribution learning, resulting in a fairer learned distribution. One limitation of our approach is that the quality of generated images as well as the resulting distribution depend on the performance of the pre-trained model. As we only "intervene" on the latent distribution of a given encoder, our performance may be limited by a noisy generator (e.g., encoding and decoding an image of a male could generate that of a female, or vice versa). Possible future work includes addressing this encoder-decoder noise, for instance through additional fine-tuning or learning the circuit to correct for such noise. This can also increase the applicability of our method to flexibly control the sensitive attribute distribution through conditional sampling, not necessarily limited to the reference data distribution.

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

Table 3: Average number of epochs to converge for the proposed method.

| F-M Ratio | 80-20 | | | | 90-10 | | | |
|---|---|---|---|---|---|---|---|---|
| $\gamma$ | 0.1 | 0.25 | 0.5 | 1.0 | 0.1 | 0.25 | 0.5 | 1.0 |
| Ours | 1365.0 | 1037.8 | 724.4 | 491.9 | 1375.6 | 1142.9 | 746.1 | 551.8 |
| Ours $+\boldsymbol{S}$ | 1103.5 | 955.4 | 636.8 | 435.8 | 1098.1 | 997.2 | 678.2 | 500.9 |

## A  APPENDIX

### A.1  TIMING

The average number of epochs to converge is provided in Table 3. For the baseline method (Choi et al., 2020a), the number of epochs is 150, and each experiment was done once.

### A.2  COMMON ALGORITHMS

The common algorithm for learning a PC with SGD-based negative likelihood is given in 2. According to the algorithm, the circuit parameters $\psi$ are updated in every iteration so that it has a greater likelihood for the input variable. You can also see how PCs generate samples in Algorithm 3.

---

**Algorithm 2** Training the PC on $\boldsymbol{Z}$ with negative log-likelihood loss

---

1: **Input:** train set $\mathcal{D}_{train}$, validation set $\mathcal{D}_{val}$, number of epochs $N_e$, tolerance $\epsilon$
2: **Output:** PC $\mathcal{C}$ with weights and leaf node parameters $\psi$
3: **for** batch $b$ **in** $\mathcal{D}_{train}$ **do**
4:     $\mathcal{LL} = PC(b)$
5:     $\mathcal{NLL} = -\mathcal{LL}$
6:     Back-propagate $\mathcal{NLL}$ to PC
7:     Update $\psi$
8: **repeat**
9:     $ValLoss$ = Compute Loss on $\mathcal{D}_{val}$
10: **until** $ValLoss$ reduced less than $\epsilon$ or $epoch = N_e$
11: **return** $\mathcal{C}$

---

---

**Algorithm 3** Sampling from Probabilistic Circuits (Dang et al., 2022)

---

1: **Input:** A PC, $\mathcal{C}$ representing a distribution $p(X)$.
2: **Output:** An instance $X$ from $\mathcal{C}$.
3: **function** SAMPLE(n)
4:     **if** n is a leaf node **then**
5:         $f_n(x) \leftarrow$ univariate distribution of $n$; **return** $x \sim f_n(x)$
6:     **else if** n is a product node **then**
7:         $x_c \leftarrow$ Sample(c) for each $c \in in(n)$; **return** Concat($\{x_c\}_{c \in in(n)}$)
8:     **else** n is a sum node
9:         sample a child unit $c$ proportional to $\{\theta_{c|n}\}_{c \in in(n)}$; **return** Sample(c)
10:     **return** Sample(r) where r is the root node

---

### A.3  MORE GENERATED SAMPLES

In this section, the generated samples are provided for different $\mathcal{D}_{bias}$ configurations. Figure 6 shows some samples from the VQ-GAN (Esser et al., 2020) transformer. The rest of the figures show samples of the proposed method for both guided and unguided settings.

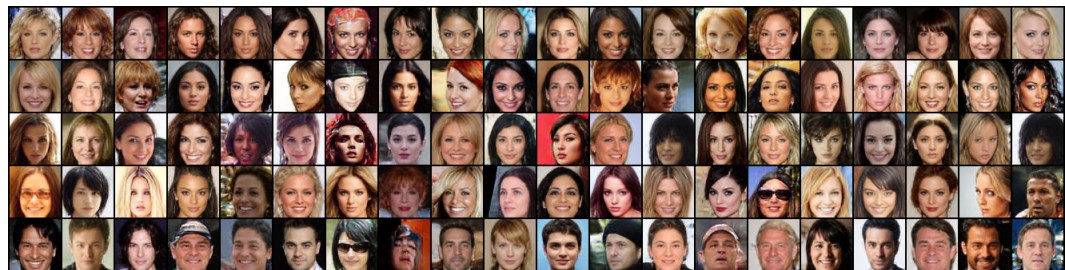

Figure 6: The generated images using VQ-GAN (Esser et al., 2020) transformer when female-male ratio in $\mathcal{D}_{bias}$ is 80-20, and $\gamma = 0.25$. For this set of generated samples, the proportion of females is 0.79 and males is 0.21

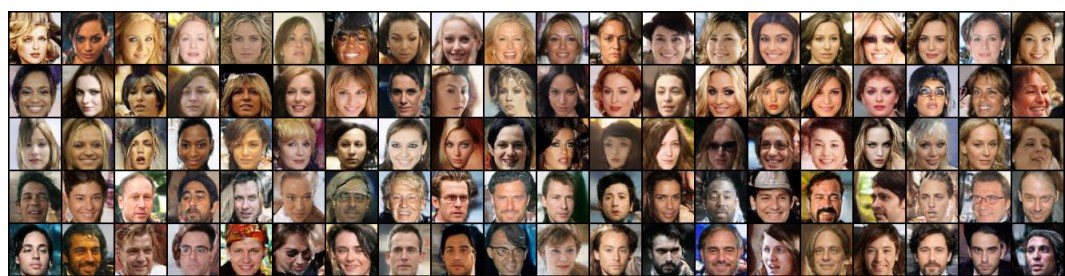

Figure 7: Generated samples by first method. (F-M Ratio 80-20, $\gamma = 0.1$)

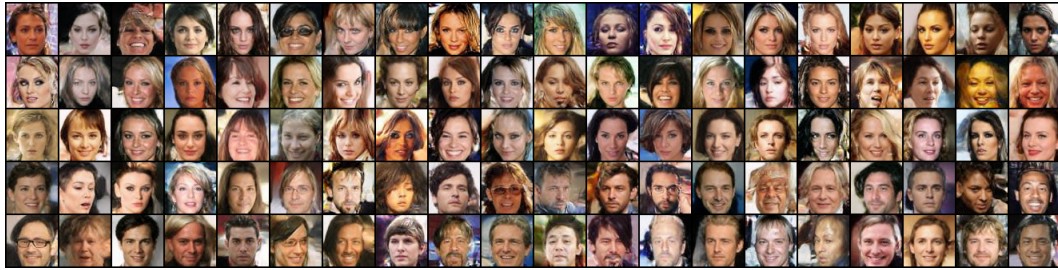

Figure 8: Generated samples by first method. (F-M Ratio 80-20, $\gamma = 0.25$)

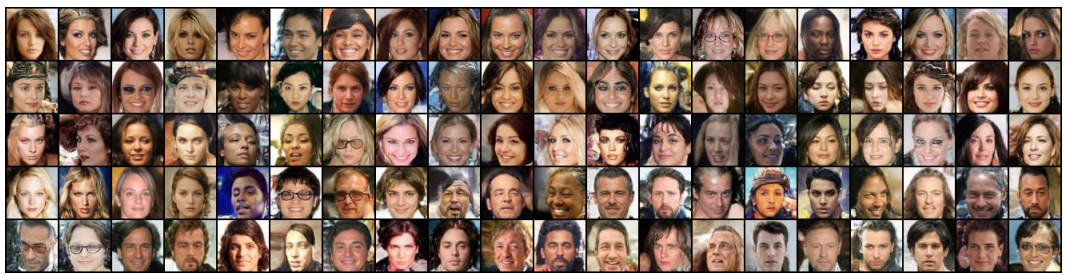

Figure 9: Generated samples by first method. (F-M Ratio 80-20, $\gamma = 0.5$)

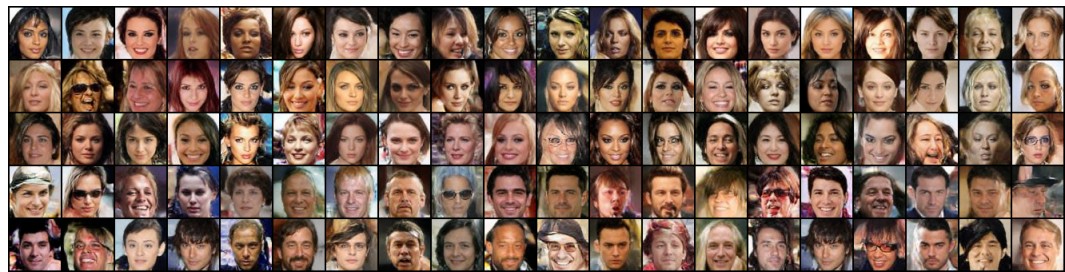

Figure 10: Generated samples by first method. (F-M Ratio 80-20, $\gamma = 1.0$)

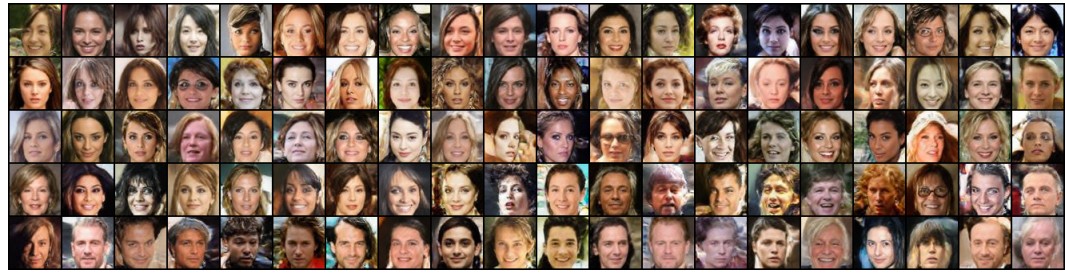

Figure 11: Generated samples by first method. (F-M Ratio 90-10, $\gamma = 0.1$)

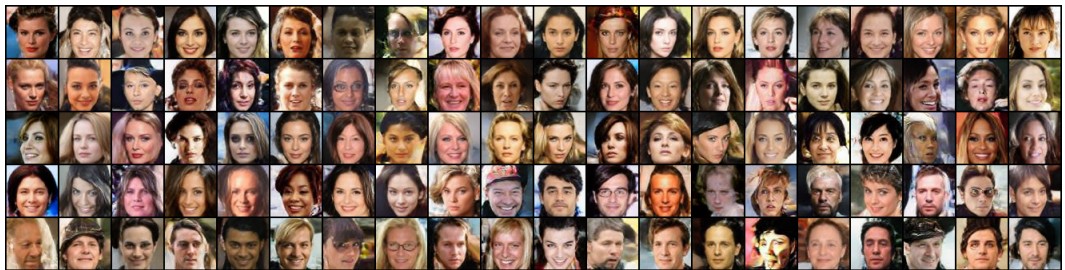

Figure 12: Generated samples by first method. (F-M Ratio 90-10, $\gamma = 0.25$)

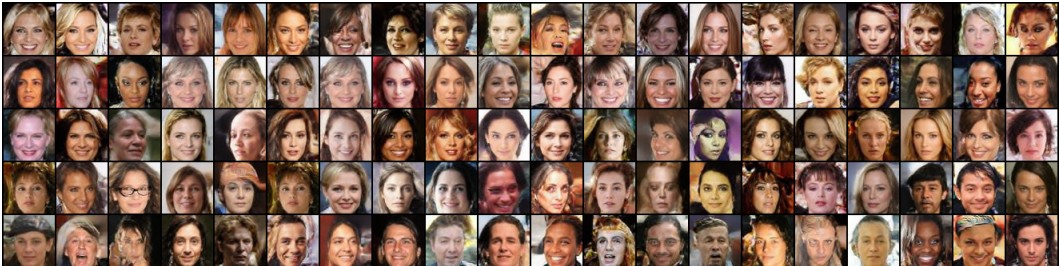

Figure 13: Generated samples by first method. (F-M Ratio 90-10, $\gamma = 0.5$)

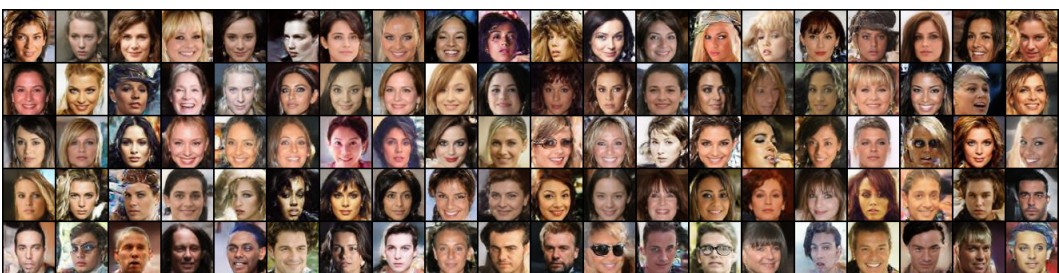

Figure 14: Generated samples by first method. (F-M Ratio 90-10, $\gamma = 1.0$)

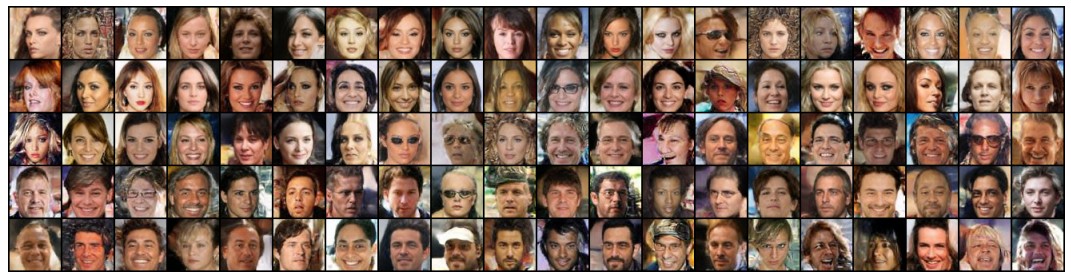

Figure 15: Generated samples by second method ($\mathbf{Z} + \mathbf{S}$). (F-M Ratio 80-20, $\gamma = 0.1$)

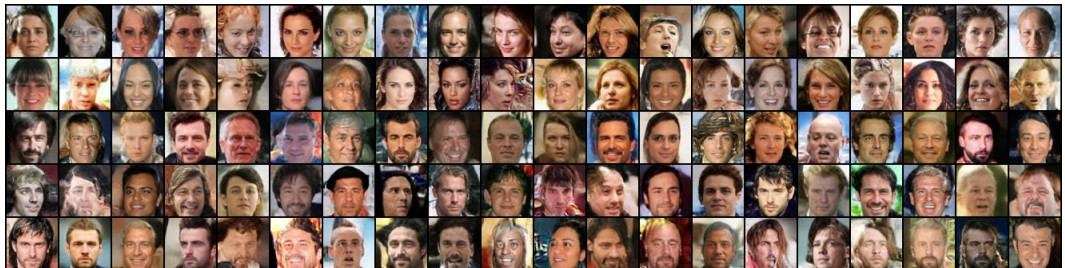

Figure 16: Generated samples by second method ($\mathbf{Z} + \mathbf{S}$). (F-M Ratio 80-20, $\gamma = 0.25$)

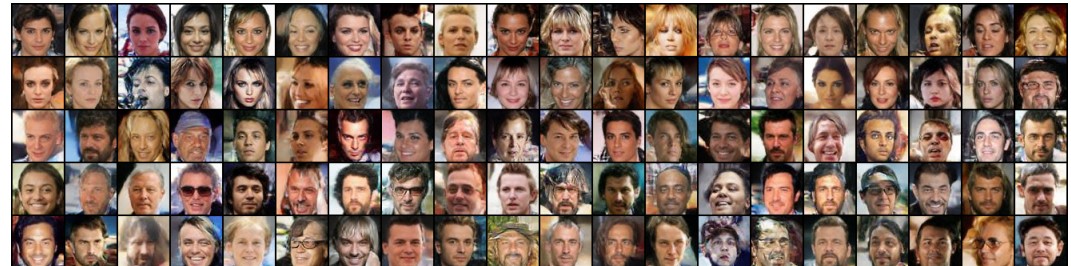

Figure 17: Generated samples by second method ($\mathbf{Z} + \mathbf{S}$). (F-M Ratio 80-20, $\gamma = 0.5$)

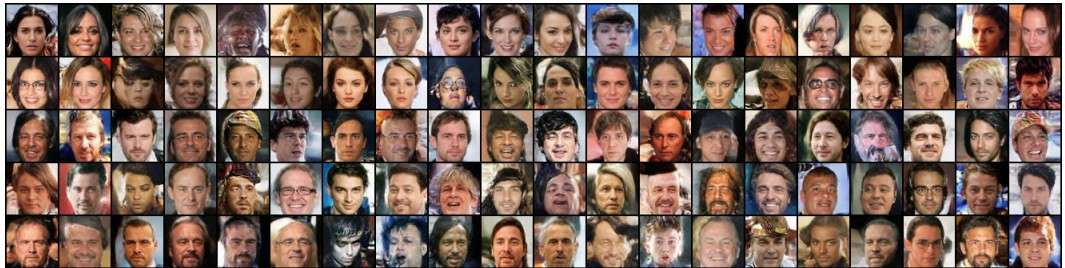

Figure 18: Generated samples by second method ($\mathbf{Z} + \mathbf{S}$). (F-M Ratio 80-20, $\gamma = 1.0$)

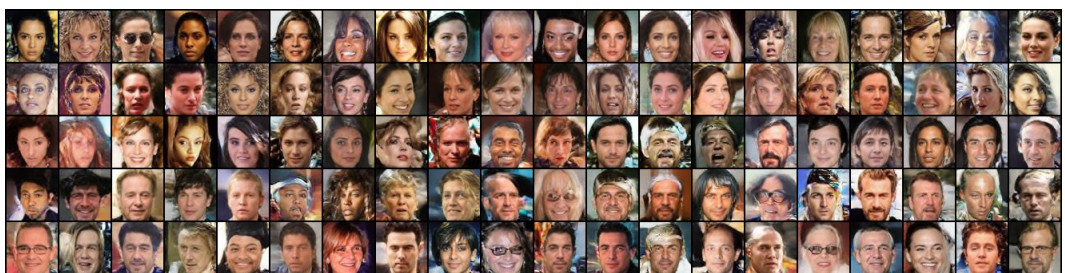

Figure 19: Generated samples by second method ($\mathbf{Z} + \mathbf{S}$). (F-M Ratio 90-10, $\gamma = 0.1$)

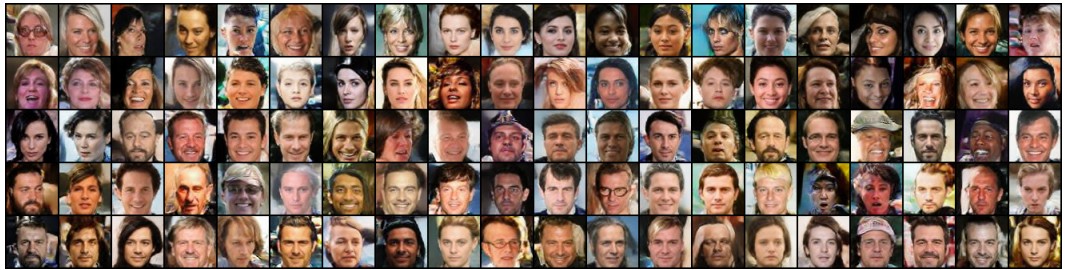

Figure 20: Generated samples by second method ($\boldsymbol{Z} + \boldsymbol{S}$). (F-M Ratio 90-10, $\gamma = 0.25$)

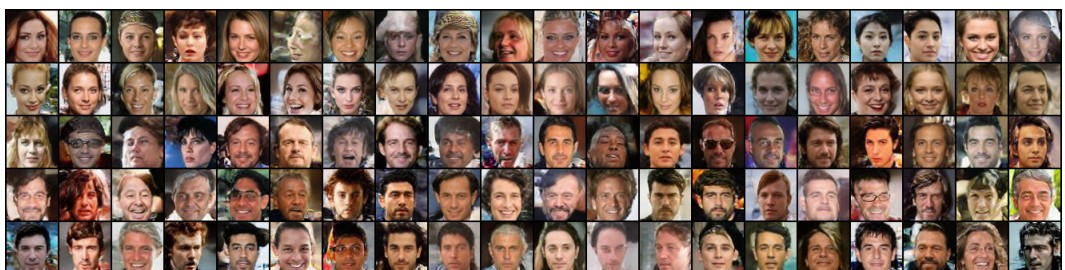

Figure 21: Generated samples by second method ($\boldsymbol{Z} + \boldsymbol{S}$). (F-M Ratio 90-10, $\gamma = 0.5$)

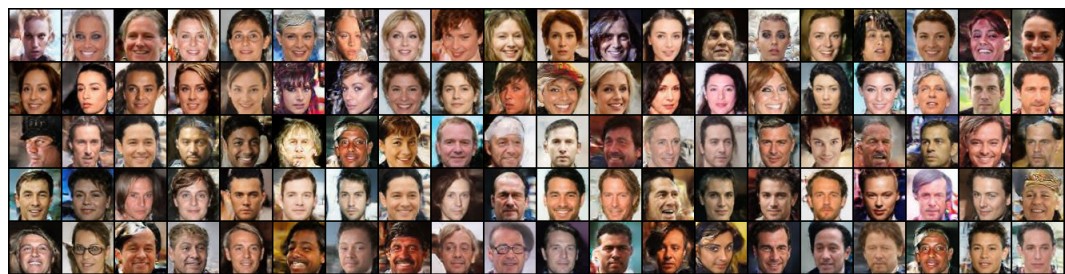

Figure 22: Generated samples by second method ($\boldsymbol{Z} + \boldsymbol{S}$). (F-M Ratio 90-10, $\gamma = 1.0$)

