# OpenReview forum: "Fair Image Generation from Pre-trained Models by Probabilistic Modeling"
_NeurIPS.cc/2024/Workshop/SafeGenAi — SafeGenAi Poster_

### Official Review · Reviewer_DYkq · 2024-10-09
**Exciting theory and experiment but future work is necessary**

**Rating:** 6
**Confidence:** 3

**Review:**

The paper presents an approach to equip probabilistic modeling to a pre-trained generative model
to fix unfair distribution without retraining.

The proposal is insightful to save the training resources when achieving fair generation. It is convincing with the experiments that the theory can enable a generation model to leverage the distribution of a small reference dataset.
Yet, the there are many results lacking analysis in Table 1, e.g.:
- A bigger size of reference dataset does not always help the guided method.
- The guided method seems to result in better fairness while the pre-trained data is more skewed.

The new model is currently only leveraging the reference data distribution, this may related to some issues and limitations:
- The experiment shows that the method is likely to be sensitive to data.
- The generation becomes noisy, i.e. the fairness is fixed at the cost of faithfulness.
- It is not clear whether the image quality will be impacted a lot if the reference data are sourced somewhere different from pre-training.
Future work like conditional sampling seems necessary.

Overall: exciting theory and experiment but future work is necessary

P.S. In Algorithm 2, train data is denoted as $D_{val}$

---

### Official Review · Reviewer_GPXH · 2024-10-09
**An Approach with Promise and Room for Growth**

**Rating:** 6
**Confidence:** 5

**Review:**

The paper is well-structured, following a standard format with clear sections for introduction, related work, background, proposed method, experiments, and conclusion. However, there are some areas for improvement:

1. The abstract provides a good overview but could be more concise and highlight the key results more clearly.
2. Some figures (e.g., Fig. 1 and Fig. 2) lack detailed captions, making it difficult to fully understand them without referring to the main text.
3. The notation section (3.1) is helpful, but some symbols are used before being defined (e.g., S in section 3.2).
4. The paper would benefit from a more detailed discussion of limitations and future work.

The paper addresses an important and timely issue in AI ethics and fairness:
1. It tackles the problem of bias in image generation, which is crucial as generative models become more prevalent in various applications.
2. The proposed method offers a novel approach to fair image generation without requiring retraining of large models, which is significant for practical applications.
3. The work bridges the gap between probabilistic modeling (using Probabilistic Circuits) and fair AI, potentially opening new research directions.


The methodology appears sound, with a few considerations:
1. The authors provide a clear theoretical foundation for their approach, grounding it in probabilistic modeling.
2. The experimental setup is well-described, allowing for reproducibility.
3. The use of multiple metrics (FD, FID, IS) to evaluate both fairness and image quality is commendable.
4. The comparison with baseline methods and ablation studies strengthen the validity of the results.
5. However, the paper could benefit from a more rigorous statistical analysis of the results, including confidence intervals or significance tests.

Paper Strengths:
1. Novel approach: The use of Probabilistic Circuits for fair image generation is innovative.
2. Efficiency: The method shows significant improvements in training time compared to baselines.
3. Flexibility: The approach can be integrated with various pre-trained generative models.
4. Data efficiency: The method performs well even with small fair reference datasets.
5. Comprehensive evaluation: The authors use multiple metrics and dataset configurations to validate their approach.

Paper Weaknesses:

1. Limited scope: The experiments focus only on the CelebA dataset and gender as the sensitive attribute. Testing on more diverse datasets and attributes would strengthen the claims.
2. Dependence on pre-trained model quality: As acknowledged by the authors, the method's performance is limited by the quality of the pre-trained encoder-decoder pair.
3. Lack of comparison with state-of-the-art fair image generation methods beyond (Choi et al., 2020a).
4. The discussion of limitations and future work is relatively brief and could be expanded.

Additional Comments and Questions:
1. How does the method perform on multi-attribute fairness scenarios (e.g., considering both gender and race)?
2. What are the computational requirements for training and inference using this method compared to baseline approaches?
3. How sensitive is the method to the choice of PC architecture and hyperparameters?
4. How does the method handle cases where the fair reference dataset is very small or potentially biased itself?
5. The paper mentions the possibility of conditional sampling but doesn't explore it fully. This could be an interesting direction for future work.
6. The authors could consider discussing how their method relates to causal approaches to fairness in machine learning.

---

### Official Review · Reviewer_5evH · 2024-10-10
**Review of Fair Image Generation by Probabilistic Modelling**

**Rating:** 6
**Confidence:** 4

**Review:**

This paper presents a novel approach to address fairness in image generation by intervening over the latent distributions of pre-trained models using probabilistic circuits without retraining the entire model.

Strengths:

The use of probabilistic circuits to learn a fair distribution and guide subsequent image generation in a way that ensures fairness is a novel approach.

The technique is highly efficient, as it doesn't require retraining the pre-trained model. Instead, it adjusts the latent space using a small fair reference dataset, saving computational resources and time.

The ability to intervene in the latent distribution and adjust the level of fairness provides users with control over the trade-off between fairness and fidelity.

Weaknesses:

The paper focuses entirely on gender as the sensitive attribute in its fairness evaluation. It would be interesting to evaluate on other datasets with multiple set of sensitive attributes, like race, age…

While the explanation of how probabilistic circuits influence latent distributions is presented, the paper does not provide theoretical guarantees on the fairness of the generated images. A formal discussion on fairness guarantees would improve the overall strength of the claims.

While the guided method improves fairness, its dependance on the availability of sensitive attribute labels may limit the applicability in cases where such labels are unavailable.

Although the method aims to balance fairness and fidelity, there is some degradation in image quality as shown by the FID scores indicating scope for improvement.

Conclusion:

The paper presents an efficient method for ensuring fairness in image generation by using probabilistic circuits to adjust the latent distributions of pre-trained models. However the lack of theoretical fairness guarantees, availability of sensitive attributes and degradation of image quality highlight the scope for improvement in future work.